# Quasi-Hyperbolically Symmetric *γ*-Metric

**DOI:** 10.3390/e25091338

**Published:** 2023-09-15

**Authors:** Luis Herrera, Alicia Di Prisco, Justo Ospino, Jaume Carot

**Affiliations:** 1Instituto Universitario de Física Fundamental y Matemáticas, Universidad de Salamanca, 37007 Salamanca, Spain; 2Escuela de Física, Facultad de Ciencias, Universidad Central de Venezuela, Caracas 1050, Venezuela; alicia.diprisco@ucv.ve; 3Departamento de Matemáticas Aplicada and Instituto Universitario de Física Fundamental y Matemáticas, Universidad de Salamanca, 37007 Salamanca, Spain; j.ospino@usal.es; 4Departament de Física, Universitat Illes Balears, 07122 Palma de Mallorca, Spain; jcarot@uib.cat

**Keywords:** black holes, exact solutions, general relativity, 04.40.-b, 04.20.-q, 04.40.Dg, 04.40.Nr

## Abstract

We carry out a systematic study on the motion of test particles in the region inner to the naked singularity of a quasi-hyperbolically symmetric γ-metric. The geodesic equations are written and analyzed in detail. The obtained results are contrasted with the corresponding results obtained for the axially symmetric γ-metric and the hyperbolically symmetric black hole. As in this latter case, it is found that test particles experience a repulsive force within the horizon (naked singularity), which prevents them from reaching the center. However, in the present case, this behavior is affected by the parameter γ which measures the departure from the hyperbolical symmetry. These results are obtained for radially moving particles as well as for particles moving in the θ−r subspace. The possible relevance of these results in the explanation of extragalactic jets is revealed.

## 1. Introduction

In a recent paper [1], an alternative global description of the Schwarzschild black hole has been proposed. The motivation behind such an endeavor is, on the one hand, the fact that the spacetime within the horizon, in the classical picture, is necessarily non-static or, in other words, that any transformation that maintains the static form of the Schwarzschild metric (in the whole spacetime) is unable to remove the coordinate singularity appearing on the horizon in the line element [2]. Indeed, as is well known, no static observers can be defined inside the horizon (see [3,4] for a discussion on this point). This conclusion becomes intelligible if we recall that the Schwarzschild horizon is also a Killing horizon, implying that the time-like Killing vector existing outside the horizon becomes space-like inside it.

On the other hand, based on the physically reasonable point of view that any equilibrium final state of a physical process should be static, it would be desirable to have a static solution over the whole spacetime.

Based on the arguments above, the following model is proposed in [1].

Outside the horizon (R>2M), one has the usual Schwarzschild line element corresponding to the spherically symmetric vacuum solution to the Einstein equations, which in polar coordinate reads (with signature +2)
(1)$$\begin{array}{ccc} ds^{2} & = & -\left(1-\frac{2M}{R}\right)dt^{2}+\frac{dR^{2}}{\left(1-\frac{2M}{R}\right)}+R^{2}d\Omega^{2},\\ d\Omega^{2} & = & d\theta^{2}+\sin^{2}\theta d\phi^{2}. \end{array}$$

This metric is static and spherically symmetric, meaning that it admits four Killing vectors:(2)$$\begin{array}{c} X_{(0)}=\partial_{t},\;X_{\left(2\right)}=-\cos\phi \partial_{\theta }+\cot\theta \sin\phi \partial_{\phi },\\ X_{\left(1\right)}=\partial_{\phi },\;X_{\left(3\right)}=\sin\phi \partial_{\theta }+\cot\theta \cos\phi \partial_{\phi }. \end{array}$$

The solution proposed for R<2M (with signature −2) is
(3)$$\begin{array}{ccc} ds^{2} & = & \left(\frac{2M}{R}-1\right)dt^{2}-\frac{dR^{2}}{\left(\frac{2M}{R}-1\right)}-R^{2}d\Omega^{2},\\ d\Omega^{2} & = & d\theta^{2}+\sinh^{2}\theta d\phi^{2}. \end{array}$$

This is a static solution, meaning that it admits the time-like Killing vector $X_{\left(0\right)}$; however, unlike (Equation 1), it is not spherically symmetric but hyperbolically symmetric, meaning that it admits the three Killing vectors
(4)$$\begin{array}{c} Y_{\left(2\right)}=-\cos\phi \partial_{\theta }+\coth\theta \sin\phi \partial_{\phi },\\ Y_{\left(1\right)}=\partial_{\phi },\;Y_{\left(3\right)}=\sin\phi \partial_{\theta }+\coth\theta \cos\phi \partial_{\phi }. \end{array}$$

Thus, if one wishes to keep sphericity within the horizon, one should abandon staticity, and if one wishes to keep staticity within the horizon, one should abandon sphericity.

The classical picture of the black hole entails sphericity within the horizon; instead, in [1], we have proceeded differently and have assumed staticity within the horizon.

The three Killing vectors (Equation 4) define the hyperbolical symmetry. Spacetimes endowed with hyperbolical symmetry have previously been the subject of research in different contexts (see [5,6,7,8,9,10,11,12,13,14,15,16,17,18,19,20,21,22,23,24,25] and the references therein).

In [13], a general study of geodesics in the spacetime described by (Equation 3) is presented (see also [20]), leading to some interesting conclusions about the behavior of a test particle in this new picture of the Schwarzschild black hole:The gravitational force inside the region R<2M is repulsive.Test particles cannot reach the center.Test particles can cross the horizon outward but only along the θ=0 axis.

These intriguing results further reinforce the interest in this kind of system.

The procedure used in [1] to obtain (Equation 3) may be used to obtain hyperbolic versions of other spacetimes. Of course, in this case, the obtained metric may not admit all the Killing vectors describing the hyperbolical symmetry (Equation 4), and it will not describe a black hole but a naked singularity. We shall refer to these spacetimes as quasi-hyperbolical.

It is the purpose of this work to delve deeper into this issue by considering a specific quasi-hyperbolical spacetime. Thus, we shall analyze the quasi-hyperbolical version of the γ-metric [26,27,28,29]. In particular, we endeavor to analyze the geodesic structure of this spacetime and to contrast it with the corresponding geodesics of the hyperbolically symmetric version of the Schwarzschild metric discussed in [13] and with the geodesic structure of the γ-metric discussed in [30].

The motivation for this choice is twofold. On the one hand, the γ-metric corresponds to a solution of the Laplace equation, in cylindrical coordinates, with the same Newtonian source image [31] as the Schwarzschild metric (a rod). On the other hand, it has been proved [32] that by extending the length of the rod to infinity one obtains the Levi-Civita spacetime. At the same time, a link was established between the parameter γ, measuring the mass density of the rod in the γ-metric, and the parameter σ, which is thought to be related to the energy density of the source of the Levi-Civita spacetime. The limit of the γ-metric when extending its rod source image to an infinite length produces, intriguingly, the flat Rindler spacetime. This result enhances even more the peculiar character of the γ-spacetime.

In other words, the γ-metric is an appealing candidate to describe spacetimes close to Schwarzschild, by means of exact analytical solutions to Einstein vacuum equations. This of course is of utmost relevance and explains why it has been so extensively studied in the past (see [33,34,35,36,37,38,39,40,41,42,43,44,45,46,47,48,49,50,51,52] and the references therein).

This line of research is further motivated by a promising new trend of investigations aimed at developing tests of gravity theories and corresponding black hole (or naked singularities) solutions for strong gravitational fields, which is based on the recent observations of shadow images of the gravitationally collapsed objects at the center of the elliptical galaxy M87 and at the center of the Milky Way galaxy by the Event Horizon Telescope (EHT) Collaboration [53,54]. The important point is that GR has not been tested yet for such strong fields [55,56,57]. The data from EHT observations can be used to obtain constraints on the parameters of the mathematical solutions that could describe the geometry surrounding those objects. These solutions include, among others, black hole spacetimes in modified and alternative theories of gravity [58,59,60,61,62], naked singularities, as well as the classical GR black hole with hair or immersed in matter fields [63,64,65,66,67,68].

Our purpose in this paper is to provide another, yet static non-spherical exact solution to vacuum Einstein equations, which could be tested against the results of the Event Horizon Telescope (EHT) Collaboration. In order to do so, we shall analyze in detail the geodesics of test particles in the field of the quasi-hyperbolical γ-metric.

## 2. The γ-Metric and Its Hyperbolic Version

In Erez–Rosen coordinates, the line element for the γ-metric is
(5)$$ds^{2}=fdt^{2}-f^{-1}\left[gdr^{2}+hd\theta^{2}+\left(r^{2}-2mr\right)\sin^{2}\theta d\phi^{2}\right],$$
where
(6)$$\begin{array}{ccc} f & = & {\left(1-\frac{2m}{r}\right)}^{\gamma }, \end{array}$$
(7)$$\begin{array}{ccc} g & = & {\left(\frac{1-\frac{2m}{r}}{1-\frac{2m}{r}+\frac{m^{2}}{r^{2}}\sin^{2}\theta }\right)}^{\gamma^{2}-1}, \end{array}$$
(8)$$\begin{array}{ccc} h & = & \frac{r^{2}{\left(1-\frac{2m}{r}\right)}^{\gamma^{2}}}{{\left(1-\frac{2m}{r}+\frac{m^{2}}{r^{2}}\sin^{2}\theta \right)}^{\gamma^{2}-1}}, \end{array}$$
and γ is a constant parameter.

The mass (monopole) *M* and the quadrupole moment *Q* of the solution are given by
(9)$$M=\gamma m,\;Q=\gamma \left(1-\gamma^{2}\right)\frac{m^{3}}{3},$$
implying that the source will be oblate (prolate) for γ>1 (γ<1). Obviously, for γ=1, we recover the Schwarzschild solution.

The hyperbolic version of (Equation 5) reads
(10)$$ds^{2}=Fdt^{2}-F^{-1}\left[Gdr^{2}+Hd\theta^{2}+\left(2mr-r^{2}\right)\sinh^{2}\theta d\phi^{2}\right],$$
where
(11)$$\begin{array}{ccc} F & = & {\left(\frac{2m}{r}-1\right)}^{\gamma }, \end{array}$$
(12)$$\begin{array}{ccc} G & = & {\left(\frac{\frac{2m}{r}-1}{\frac{2m}{r}-1+\frac{m^{2}}{r^{2}}\sinh^{2}\theta }\right)}^{\gamma^{2}-1}, \end{array}$$
(13)$$\begin{array}{ccc} H & = & \frac{r^{2}{\left(\frac{2m}{r}-1\right)}^{\gamma^{2}}}{{\left(\frac{2m}{r}-1+\frac{m^{2}}{r^{2}}\sinh^{2}\theta \right)}^{\gamma^{2}-1}}, \end{array}$$
which can be very easily obtained by following the procedure used in [1] to obtain (Equation 3) from (Equation 1). It is easy to check that (Equation 10) is a solution to vacuum Einstein equations and that γ=1 corresponds to the line element (Equation 3).

Thus, as in [1], we shall assume that the line element defined by (Equation 10) describes the region r<2m, whereas the spacetime outside r=2m is described by the “usual” γ-metric (Equation 5). However, in this case, if γ≠1, the surface r=2m represents a naked singularity because the curvature invariants are singular on that surface (as expected from the Israel theorem [69]).

Indeed, the calculation of the Kretschmann scalar K
(14)$$K=R_{\alpha \beta \mu \nu }R^{\alpha \beta \mu \nu },$$
for (Equation 10), produces
(15)$$\begin{array}{ccc} K & = & \frac{64m^{2}\gamma^{2}{\left(\frac{2m}{r}-1\right)}^{2\gamma }{\left(1+\frac{m^{2}\sinh^{2}\theta }{2mr-r^{2}}\right)}^{2\gamma^{2}}}{r^{2}{\left(-2m+r\right)}^{2}{\left(-m^{2}+4mr-2r^{2}+m^{2}\cosh2\theta \right)}^{3}}\left\{-6r^{4}+12mr^{3}\left(2+\gamma \right)\right.\\ & + & \left.3m^{3}r{\left(1+\gamma \right)}^{2}\left(4+\gamma \right)-m^{4}{\left(1+\gamma \right)}^{2}\left(1+\gamma +\gamma^{2}\right)\left(1-\cosh2\theta \right)\right.\\ & & \left.-3m^{2}r^{2}\left[10+3\gamma \left(4+\gamma \right)\right]+m^{2}\left[3r^{2}\gamma^{2}-3mr\gamma {\left(1+\gamma \right)}^{2}\right]\cosh2\theta \right\}, \end{array}$$
which is singular at r=2m, except for γ=1, in which case we obtain
(16)$$K=\frac{48m^{2}}{r^{6}}.$$

As it is evident that the metric (Equation 10) does not admit the three Killing vectors (Equation 4), the γ-metric (Equation 5) does not admit the Killing vectors (Equation 2) describing the spherical symmetry as well.

Indeed, from
(17)$$L_{X}g_{\alpha \beta }=X^{\rho }\partial_{\rho }g_{\alpha \beta }+g_{\alpha \rho }\partial_{\beta }X^{\rho }+g_{\beta \rho }\partial_{\alpha }X^{\rho },$$
where $L_{X}$ denotes the Lie derivative with respect to the vectors (Equation 2), we obtain for (Equation 5) two non-vanishing independent components of (Equation 17)
(18)$$L_{X}g_{\alpha \beta }K^{\alpha }K^{\beta }=L_{X}g_{\alpha \beta }L^{\alpha }L^{\beta }=\frac{m^{2}\left(1-\gamma^{2}\right)\sin2\theta \cos\phi }{r^{2}\left(1-\frac{2m}{r}+\frac{m^{2}}{r^{2}}\sin^{2}\theta \right)},$$
(19)$$\begin{array}{c} L_{X}g_{\alpha \beta }L^{\alpha }S^{\beta }=-\frac{\sin\phi }{\sin\theta }\left[{\left(\frac{1-\frac{2m}{r}}{1-\frac{2m}{r}+\frac{m^{2}}{r^{2}}\sin^{2}\theta }\right)}^{\frac{\gamma^{2}-1}{2}}\right.\\ \left.-{\left(\frac{1-\frac{2m}{r}+\frac{m^{2}}{r^{2}}\sin^{2}\theta }{1-\frac{2m}{r}}\right)}^{\frac{\gamma^{2}-1}{2}}\right], \end{array}$$
where the orthogonal tetrad associated to (Equation 5) is
$$\begin{array}{ccc} V^{\alpha } & = & \left(\frac{1}{\sqrt{f}},0,0,0\right),\;K^{\alpha }=\left(0,\sqrt{f/g},0,0\right),\\ L^{\alpha } & = & \left(0,0,\sqrt{f/h},0\right),\;S^{\alpha }=\left(0,0,0,\frac{\sqrt{f}}{r\sin\theta \sqrt{1-\frac{2m}{r}}}\right). \end{array}$$

On the other hand, calculating $L_{Y}g_{\alpha \beta }$ for (Equation 10) and (Equation 4), we obtain two non-vanishing components
(20)$$L_{Y}g_{\alpha \beta }{\tilde{K}}^{\alpha }{\tilde{K}}^{\beta }=L_{\chi }{\tilde{g}}_{\alpha \beta }{\tilde{L}}^{\alpha }{\tilde{L}}^{\beta }=\frac{m^{2}\left(\gamma^{2}-1\right)\sinh2\theta \cos\phi }{r^{2}\left(\frac{2m}{r}-1+\frac{m^{2}}{r^{2}}\sinh^{2}\theta \right)},$$
(21)$$\begin{array}{c} L_{Y}g_{\alpha \beta }{\tilde{L}}^{\alpha }{\tilde{S}}^{\beta }=\frac{\sin\phi }{\sinh\theta }\left[{\left(\frac{\frac{2m}{r}-1}{\frac{2m}{r}-1+\frac{m^{2}}{r^{2}}\sinh^{2}\theta }\right)}^{\frac{\gamma^{2}-1}{2}}\right.\\ \left.-{\left(\frac{\frac{2m}{r}-1+\frac{m^{2}}{r^{2}}\sinh^{2}\theta }{\frac{2m}{r}-1}\right)}^{\frac{\gamma^{2}-1}{2}}\right], \end{array}$$
where the orthogonal tetrad associated to (Equation 10) is
$$\begin{array}{ccc} {\tilde{V}}^{\alpha } & = & \left(\frac{1}{\sqrt{F}},0,0,0\right),\;{\tilde{K}}^{\alpha }=\left(0,\sqrt{F/G},0,0\right),\\ {\tilde{L}}^{\alpha } & = & \left(0,0,\sqrt{F/H},0\right),\;{\tilde{S}}^{\alpha }=\left(0,0,0,\frac{\sqrt{F}}{r\sinh\theta \sqrt{\frac{2m}{r}-1}}\right). \end{array}$$

In other words, the γ-metric deviates from spherical symmetry in a similar way as the hyperbolic version of the γ-metric deviates from hyperbolical symmetry. This is the origin of the term “quasi–hyperbolically symmetric” applied to (Equation 10).

## 3. Geodesics

We shall now find the geodesic equations for test particles in the metric (Equation 10). The qualitative differences in the trajectories of the test particles as compared with the γ-metric and the metric (Equation 3) will be identified and discussed.

The equations governing the geodesics can be derived from the Lagrangian
(22)$$2L=g_{\alpha \beta }{\dot{x}}^{\alpha }{\dot{x}}^{\beta },$$
where the dot denotes differentiation with respect to an affine parameter *s*, which for time-like geodesics coincides with the proper time.

Then, from the Euler–Lagrange equations,
(23)$$\frac{d}{ds}\left(\frac{\partial L}{\partial {\dot{x}}^{\alpha }}\right)-\frac{\partial L}{\partial x^{\alpha }}=0,$$
we obtain for (Equation 10)
(24)$$\ddot{t}-\frac{2\gamma m}{r^{2}\left(\frac{2m}{r}-1\right)}\dot{r}\dot{t}=0,$$
(25)$$\begin{array}{c} \ddot{r}-\frac{m\gamma {\left(\frac{2m}{r}-1\right)}^{2\gamma -\gamma^{2}}}{r^{2}{\left(\frac{2m}{r}-1+\frac{m^{2}}{r^{2}}\sinh^{2}\theta \right)}^{1-\gamma^{2}}}{\dot{t}}^{2}\\ -\frac{m}{r^{2}}\left[\frac{\left(\gamma^{2}-\gamma -1\right)}{\frac{2m}{r}-1}-\frac{\left(\gamma^{2}-1\right)\left(1+\frac{m}{r}\sinh^{2}\theta \right)}{\frac{2m}{r}-1+\frac{m^{2}}{r^{2}}\sinh^{2}\theta }\right]{\dot{r}}^{2}\\ -\frac{m^{2}\left(\gamma^{2}-1\right)\sinh2\theta }{r^{2}\left(\frac{2m}{r}-1+\frac{m^{2}}{r^{2}}\sinh^{2}\theta \right)}\dot{\theta }\dot{r}+\left[r+m(\gamma^{2}-\gamma -2)\right.\\ \left.-\frac{m\left(\gamma^{2}-1\right)\left(1+\frac{m}{r}\sinh^{2}\theta \right)\left(\frac{2m}{r}-1\right)}{\left(\frac{2m}{r}-1+\frac{m^{2}}{r^{2}}\sinh^{2}\theta \right)}\right]{\dot{\theta }}^{2}\\ -\frac{\left[m\left(1+\gamma \right)-r\right]{\left(\frac{2m}{r}-1+\frac{m^{2}}{r^{2}}\sinh^{2}\theta \right)}^{\gamma^{2}-1}\sinh^{2}\theta }{{\left(\frac{2m}{r}-1\right)}^{\gamma^{2}-1}}{\dot{\phi }}^{2}\\ =0, \end{array}$$
(26)$$\begin{array}{ccc} & & \ddot{\theta }+\frac{m^{2}\left(\gamma^{2}-1\right)\sinh2\theta }{2r^{4}\left(\frac{2m}{r}-1\right)\left(\frac{2m}{r}-1+\frac{m^{2}}{r^{2}}\sinh^{2}\theta \right)}{\dot{r}}^{2}\\ & + & 2\left[\frac{1}{r}+\frac{m(\gamma -\gamma^{2})}{r^{2}\left(\frac{2m}{r}-1\right)}+\frac{m\left(\gamma^{2}-1\right)\left(1+\frac{m}{r}\sinh^{2}\theta \right)}{r^{2}\left(\frac{2m}{r}-1+\frac{m^{2}}{r^{2}}\sinh^{2}\theta \right)}\right]\dot{\theta }\dot{r}\\ & - & \frac{m^{2}\left(\gamma^{2}-1\right)\sinh2\theta }{2r^{2}\left(\frac{2m}{r}-1+\frac{m^{2}}{r^{2}}\sinh^{2}\theta \right)}{\dot{\theta }}^{2}\\ & - & \frac{{\left(\frac{2m}{r}-1\right)}^{1-\gamma^{2}}\sinh2\theta }{2{\left(\frac{2m}{r}-1+\frac{m^{2}}{r^{2}}\sinh^{2}\theta \right)}^{1-\gamma^{2}}}{\dot{\phi }}^{2}=0, \end{array}$$
(27)$$\ddot{\phi }+\frac{2}{r^{2}\left(\frac{2m}{r}-1\right)}\left[m\left(1+\gamma \right)-r\right]\dot{r}\dot{\phi }+\left(2\coth\theta \right)\dot{\theta }\dot{\phi }=0.$$

Let us first analyze some particular cases from which some important general results on the geodesic structure of the system may be deduced.

Thus, let us assume that at some given initial $s=s_{0}$ we have $\dot{\theta }=0$, and then it follows at once from (Equation 26) that such a condition will propagate in time only if θ=0. In other words, any θ=constant trajectory is unstable except θ=0. It is worth stressing the difference between this case and the situation in the purely hyperbolic metric where $\dot{\phi }=0$ also ensures stability.

Next, let us consider the case of circular orbits. These are defined by $\dot{r}=\dot{\theta }=0$, producing
(28)$$\begin{array}{ccc} \ddot{t}=\ddot{\phi } & = & 0, \end{array}$$
(29)$$\begin{array}{ccc} m\gamma {\dot{t}}^{2}+\frac{r^{2}\left[(\gamma +1)m-r\right]}{{\left(\frac{2m}{r}-1\right)}^{2\gamma -1}}\sinh^{2}\theta {\dot{\phi }}^{2} & = & 0, \end{array}$$
(30)$$\begin{array}{ccc} \sinh\theta \cosh\theta {\dot{\phi }}^{2} & = & 0. \end{array}$$

From (Equation 30), it is obvious that, as for the hyperbolically symmetric black hole, no circular geodesics exist in this case, which is at variance with the γ-metric spacetime.

Let us now consider the motion of a test particle along a meridional line θ ($\dot{r}=\dot{\phi }=0$). In this case, as shown in [13], motion is forbidden if γ=1; however, from (Equation 25), it is a simple matter to see that for γ>1 there are possible solutions.

More so, let us assume (always in the purely meridional motion case) that at s=0 we have θ=constant≠0 and $\dot{\theta }=0$. Then, if γ=1, it follows from (Equation 26) that $\ddot{\theta }=0$. The particle remains on the same plane, a result already obtained in [13]. However, if γ≠1, $\ddot{\theta }$ does not need to vanish, and the particle leaves the plane (θ=constant).

This effect implies the existence of a force parallel to the axis of symmetry, a result similar to the one obtained for the γ-metric, and which illustrates further the influence of the deviation from the hyperbolically symmetric case.

Let us consider the case of purely radial geodesics described by $\dot{\theta }=\dot{\phi }=0$, producing
(31)$$\ddot{t}-\frac{2\gamma m}{r^{2}\left(\frac{2m}{r}-1\right)}\dot{r}\dot{t}=0,$$
(32)$$\begin{array}{c} \ddot{r}-\frac{m\gamma {\left(\frac{2m}{r}-1\right)}^{2\gamma -\gamma^{2}}}{r^{2}{\left(\frac{2m}{r}-1+\frac{m^{2}}{r^{2}}\sinh^{2}\theta \right)}^{1-\gamma^{2}}}{\dot{t}}^{2}\\ -\frac{m}{r^{2}}\left[\frac{\left(\gamma^{2}-\gamma -1\right)}{\frac{2m}{r}-1}-\frac{\left(\gamma^{2}-1\right)\left(1+\frac{m}{r}\sinh^{2}\theta \right)}{\frac{2m}{r}-1+\frac{m^{2}}{r^{2}}\sinh^{2}\theta }\right]{\dot{r}}^{2}=0, \end{array}$$
(33)$$\frac{m^{2}\left(\gamma^{2}-1\right)\sinh2\theta }{2r^{4}\left(\frac{2m}{r}-1\right)\left(\frac{2m}{r}-1+\frac{m^{2}}{r^{2}}\sinh^{2}\theta \right)}{\dot{r}}^{2}=0.$$

The last of the equations above indicates that, if γ≠1, purely radial geodesics only exist along the axis θ=0.

In this case, it follows from (Equation 23), due to the symmetry imposed, that
(34)$$\frac{\partial L}{\partial \dot{t}}=constant=E=\dot{t}{\left(\frac{2m}{r}-1\right)}^{\gamma },$$
(35)$$\frac{\partial L}{\partial \dot{\phi }}=constant=L=-\dot{\phi }{\left(\frac{2m}{r}-1\right)}^{1-\gamma }r^{2}\sinh^{2}\theta ,$$
where *E* and *L* represent, respectively, the total energy and the angular momentum of the test particle. Because we have already seen that the only stable radial trajectory is θ=0, the angular momentum vanishes for those trajectories.

Then, using (Equation 34), we obtain for the first integral of (Equation 32)
(36)$${\dot{r}}^{2}=E^{2}-V^{2},$$
where *V*, which can be associated with the potential energy of the test particle, is given by
(37)$$V^{2}\equiv {\left(\frac{2m}{r}-1\right)}^{\gamma },$$
or, introducing the dimensionless variable x≡r/m, (Equation 37) becomes
(38)$$V^{2}={\left(\frac{2}{x}-1\right)}^{\gamma }.$$

As we see from Figure 1, for any given value of *E* (however large, but finite), the test particle inside the naked singularity never reaches the center, moving between the closest point to the center where E=V, and x=∞ because nothing prevents the particle from crossing the naked singularity outwardly. It is possible however, because for x>2 the spacetime is no longer described by (Equation 10) but by the usual γ-metric (Equation 5), that for some value of *E* the particle bounces back at a point (x>2) where E=V.

Thus, for this particular value of energy, we have a bounded trajectory with extreme points at both sides of the naked singularity. For sufficiently large (but finite) values of energy, the trajectory is unbounded and the particle moves between a point close to, but at a finite distance from, the center and r→∞.

The above picture is quite different from the behavior of the test particle in the γ-metric as described in [30] and similar to the one observed for a radially moving test particle inside the horizon for the metric (Equation 3). However, in our case, the parameter γ affects the behavior of the test particle as is apparent from Figure 1. Specifically, for γ>1, the test particle is repelled more strongly from the center, bouncing back at values of *r* larger than in the case γ≤1.

In order to understand the results above, it is convenient to calculate the four-acceleration of a static observer in the frame of (Equation 10). We recall that a static observer is one whose four-velocity $U^{\mu }$ is proportional to the Killing time-like vector [3], i.e.,
(39)$$U^{\mu }=\left[\frac{1}{{\left(\frac{2m}{r}-1\right)}^{\gamma /2}},0,0,0\right].$$

Then, for the four-acceleration $a^{\mu }\equiv U^{\beta }U_{;\beta }^{\mu }$, we obtain for the region inner to the naked singularity
(40)$$a^{\mu }=\left[0,-\frac{m\gamma {\left(\frac{2m}{r}-1+\frac{m^{2}}{r^{2}}\sinh^{2}\theta \right)}^{\gamma^{2}-1}}{r^{2}{\left(\frac{2m}{r}-1\right)}^{\gamma^{2}-\gamma }},0,0\right],$$
whereas for the region outside the naked singularity, described by (Equation 5), we obtain, with
(41)$$U^{\mu }=\left[\frac{1}{{\left(1-\frac{2m}{r}\right)}^{\gamma /2}},0,0,0\right],$$
(42)$$a^{\mu }=\left[0,\frac{m\gamma {\left(1-\frac{2m}{r}+\frac{m^{2}}{r^{2}}\sin^{2}\theta \right)}^{\gamma^{2}-1}}{r^{2}{\left(1-\frac{2m}{r}\right)}^{\gamma^{2}-\gamma }},0,0\right].$$

The physical meaning of (Equation 40) and (Equation 42) is clear; it represents the inertial radial acceleration, which is necessary in order to maintain a static frame, by canceling the gravitational acceleration exerted on the frame, for the spacetimes (Equation 10) and (Equation 5), respectively. Because this acceleration is directed radially inward (outward), in the region inner (outer) to the naked singularity, it means that the gravitational force is repulsive (attractive). The attractive nature of gravitation in (Equation 5) is expected, whereas its repulsive nature in (Equation 10) is characteristic of hyperbolical spacetimes and explains the peculiarities of the orbits inside the horizon. In particular, we see from (Equation 40) that the absolute value of the radial acceleration grows with γ, implying that the repulsion is stronger for a larger γ, as it follows from Figure 1.

We shall next consider the geodesics in the θ−r plane (ϕ=constant). The interest in this case becomes intelligible if we recall that our spacetime (Equation 10) is axially symmetric, implying that the general properties of motion on any slice ϕ=constant would be invariant with respect to rotation around the symmetry axis.

In this case, geodesic equations read
(43)$$\ddot{t}-\frac{2\gamma m}{r^{2}\left(\frac{2m}{r}-1\right)}\dot{r}\dot{t}=0,$$
(44)$$\begin{array}{c} \ddot{r}-\frac{m\gamma {\left(\frac{2m}{r}-1\right)}^{2\gamma -\gamma^{2}}}{r^{2}{\left(\frac{2m}{r}-1+\frac{m^{2}}{r^{2}}\sinh^{2}\theta \right)}^{1-\gamma^{2}}}{\dot{t}}^{2}\\ -\frac{m}{r^{2}}\left[\frac{\left(\gamma^{2}-\gamma -1\right)}{\frac{2m}{r}-1}-\frac{\left(\gamma^{2}-1\right)\left(1+\frac{m}{r}\sinh^{2}\theta \right)}{\frac{2m}{r}-1+\frac{m^{2}}{r^{2}}\sinh^{2}\theta }\right]{\dot{r}}^{2}\\ -\frac{m^{2}\left(\gamma^{2}-1\right)\sinh2\theta }{r^{2}\left(\frac{2m}{r}-1+\frac{m^{2}}{r^{2}}\sinh^{2}\theta \right)}\dot{\theta }\dot{r}+\left[r+m(\gamma^{2}-\gamma -2)\right.\\ \left.-\frac{m\left(\gamma^{2}-1\right)\left(1+\frac{m}{r}\sinh^{2}\theta \right)\left(\frac{2m}{r}-1\right)}{\left(\frac{2m}{r}-1+\frac{m^{2}}{r^{2}}\sinh^{2}\theta \right)}\right]{\dot{\theta }}^{2}=0, \end{array}$$
(45)$$\begin{array}{ccc} & & \ddot{\theta }+\frac{m^{2}\left(\gamma^{2}-1\right)\sinh2\theta }{2r^{4}\left(\frac{2m}{r}-1\right)\left(\frac{2m}{r}-1+\frac{m^{2}}{r^{2}}\sinh^{2}\theta \right)}{\dot{r}}^{2}\\ & + & 2\left[\frac{1}{r}+\frac{m(\gamma -\gamma^{2})}{r^{2}\left(\frac{2m}{r}-1\right)}+\frac{m\left(\gamma^{2}-1\right)\left(1+\frac{m}{r}\sinh^{2}\theta \right)}{r^{2}\left(\frac{2m}{r}-1+\frac{m^{2}}{r^{2}}\sinh^{2}\theta \right)}\right]\dot{\theta }\dot{r}\\ & - & \frac{m^{2}\left(\gamma^{2}-1\right)\sinh2\theta }{2r^{2}\left(\frac{2m}{r}-1+\frac{m^{2}}{r^{2}}\sinh^{2}\theta \right)}{\dot{\theta }}^{2}=0. \end{array}$$

To simplify the calculations, we shall adopt a perturbative approach assuming γ=1+ϵ, for ϵ<<1, and neglecting terms of order $\epsilon^{2}$ and higher. In doing so, we obtain from (Equation 45) at order O(0) and O(ϵ), respectively,
(46)$$(\dot{\theta }r^{2}\dot{)}=0\;\Rightarrow \;\dot{\theta }=\frac{c_{1}}{r^{2}},$$
and
(47)$$\begin{array}{ccc} & & \frac{m^{2}\sinh2\theta \;{\dot{r}}^{2}}{r^{4}\left(\frac{2m}{r}-1\right)\left(\frac{2m}{r}-1+\frac{m^{2}}{r^{2}}\sinh^{2}\theta \right)}\\ & - & \frac{m^{2}\sinh2\theta \;{\dot{\theta }}^{2}}{r^{2}\left(\frac{2m}{r}-1+\frac{m^{2}}{r^{2}}\sinh^{2}\theta \right)}\\ & - & \frac{2m}{r^{2}}\left[\frac{1-\frac{2m}{r}+\left(\frac{2m}{r}-\frac{3m^{2}}{r^{2}}\right)\sinh^{2}\theta }{\left(\frac{2m}{r}-1\right)\left(\frac{2m}{r}-1+\frac{m^{2}}{r^{2}}\sinh^{2}\theta \right)}\right]\dot{\theta }\;\dot{r}=0. \end{array}$$

Introducing
(48)$$\dot{r}=r_{\theta }\dot{\theta },\;y=\frac{m}{r},$$
Equation (Equation 47) becomes
(49)$$y_{\theta }^{2}+\frac{2y_{\theta }}{\sinh2\theta }\left[1-2y+\left(2y-3y^{2}\right)\sinh^{2}\theta \right]-y^{2}\left(2y-1\right)=0,$$
whose integration produces
(50)y=constant=1/2.

The order O(0) can be easily calculated from (Equation 25) and (Equation 46), producing
(51)$$\ddot{r}-\frac{mE^{2}}{r^{2}\left(\frac{2m}{r}-1\right)}+\frac{m\;{\dot{r}}^{2}}{r^{2}\left(\frac{2m}{r}-1\right)}+\frac{\left(r-2m\right)c_{1}^{2}}{r^{4}}=0,$$
whose first integral reads
(52)$$\dot{r}=\sqrt{E^{2}-\left(\frac{2m}{r}-1\right)\left(\frac{c_{1}^{2}}{r^{2}}+1\right)},$$
or, introducing the variable *z*
(53)$$\dot{r}=r_{\theta }\dot{\theta },\;z\equiv 2y=\frac{2m}{r},$$
(54)$$z_{\theta }=\frac{1}{k}\sqrt{E^{2}-\left(z-1\right)\left(k^{2}z^{2}+1\right)},$$
with $c_{1}=-2mk$.

This equation was already obtained and solved for the case γ=1 (Equation (38) in [13]), with the boundary condition that all trajectories coincide at θ=0,z=1. Here, we present the integration of such an equation for the values indicated in Figure 2 (please notice that for this figure we have used the variable z=2y in order to keep the same notation for the order O(0) as in [13]).

Let us now analyze in some detail the physical implications of Figure 2 and Equation (Equation 50). As we can see, the solution of the order 0(ϵ) maintains a constant value of *y* which is the same value assumed in the boundary condition. At order O(0), we see from Figure 2 that the particle never reaches the center, which may only happen as *k* and *E* tend to infinity. In either case, the particle never outwardly crosses the surface y=1/2 (z=1), thus happening only along the radial geodesic θ=0. The influence of γ in the final picture can be deduced by combining Figure 2 and Equation (Equation 50).

## 4. Discussion and Conclusions

Motivated by the relevance of the γ-metric (Equation 5) and the hyperbolically symmetric metric (Equation 3), in this work we have proposed to analyze the physical properties of the hyperbolical version of the γ-metric. Such a spacetime described by the line element (Equation 10) shares some important features with the hyperbolically symmetric spacetime described by (Equation 3), the most relevant of which is the repulsive character of gravity inside the surface r=2m. On the other hand, as for the γ-metric (Equation 5), the surface r=2m is not regular, thereby describing a naked singularity. The spacetime (Equation 10) is not hyperbolically symmetric in the sense that it does not admit the Killing vectors (Equation 4), a fact suggesting the name “quasi–hyperbolically symmetric” for such a spacetime.

We have focused our study on the characteristics of the motion of test particles in the spacetime described by (Equation 10), with special attention paid to the role of the parameter γ. Thus, our main conclusions are as follows:1.The test particles may cross the surface r=2m outwardly but only along the axis θ=0. This situation appears in the study of the geodesics in (Equation 3) presented in [13]; however, in our case, the distinctive repulsive force of this spacetime is increased by the parameter γ.2.Like in the hyperbolically symmetric case, the test particles never reach the center; however, in our case, the test particles radially directed to the center bounce back farther from the center as γ increases. This result becomes intelligible from a simple inspection of (Equation 40).3.The motion of the test particles on any slice ϕ=constant, though qualitatively similar to the case γ=1, is affected by the value of γ as follows from the analysis of Figure 2 and (Equation 50).

As we mentioned before, a new line of investigations based on observations of the shadow images of the gravitationally collapsed objects aiming to test gravity theories and the corresponding black hole (or naked singularities) solutions for strong gravitational fields is right now attracting the interest of many researchers. Such studies are particularly suitable for contrasting the physical relevance of different exact solutions to the field equations. We believe that the metric exhibited here deserves to be considered as a suitable candidate for such comparative studies. However, it is worth mentioning that we have restricted our study to time-like geodesics, whereas any contrast with ETH observational data would require results obtained from the study of null geodesics. Notwithstanding, the results obtained for time-like geodesics here presented point to the potential of the metric under consideration.

We would like to conclude with a mention to what we believe is one of the most promising applications of hyperbolical metrics. We have in mind the modeling of extragalactic relativistic jets. It should be clear that, at present, such an application remains within the realm of speculation; however, the comments below justify our (moderate) optimism.

Relativistic jets are highly energetic phenomena which have been observed in many systems (see [70,71,72,73] and the references therein), usually associated with the presence of a compact object and exhibiting a high degree of collimation. Since no consensus has been reached until now, concerning the basic mechanism explaining these two features of jets (collimation and high energies), we feel motivated to speculate that the metric here considered could be considered as a possible engine behind the jets.

Indeed, on the one hand, the collimation is ensured by the fact that test particles may outwardly cross the naked singularity but only along the θ=0 axis. On the other hand, as implied by (Equation 40), the strength of the repulsive gravitational force acting on the particle as r→0 increases as $\frac{1}{r^{(\gamma +2})}$. This explains the high energies of particles bouncing back from regions close to r=0. More so, the fact that the repulsive force would be larger for larger values of γ further enhances the efficiency of our model as the engine of such jets, as compared with the γ=1 case.

It goes without saying that confirmation of this mechanism requires a much more detailed setup based on astronomical observations of jets, which is clearly out of the scope of this work.

## Figures and Tables

**Figure 1 entropy-25-01338-f001:**
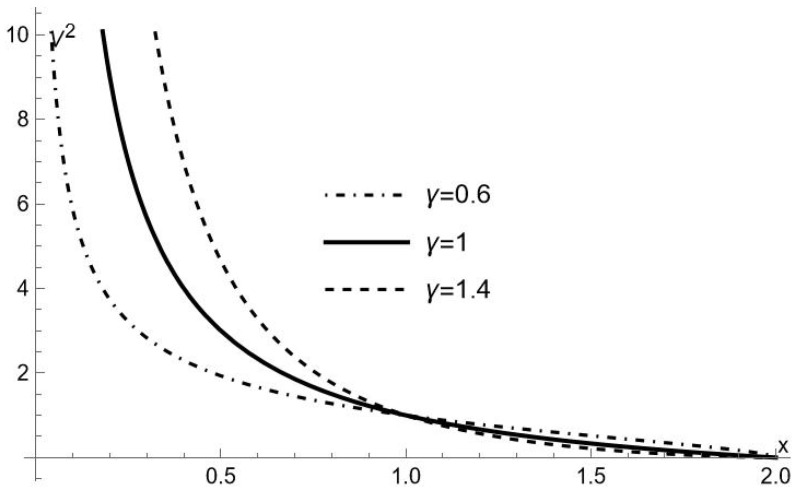
$V^{2}$ as function of *x*, for the three values of γ indicated in the figure.

**Figure 2 entropy-25-01338-f002:**
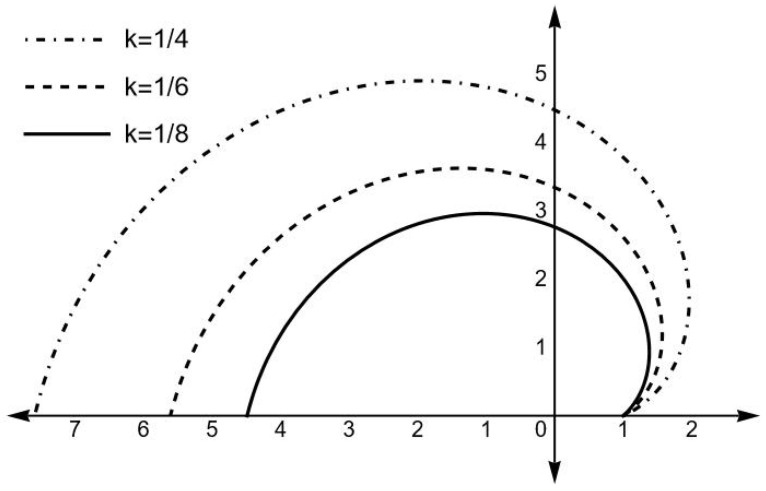
$z\equiv \frac{2m}{r}$ as function of θ, for the values of *k* indicated on the figure and E=3.

## Data Availability

Not applicable.

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
