# Peer review of "Quasi-Hyperbolically Symmetric γ-Metric"

_entropy, 2023, doi:10.3390/e25091338_

Round 1
Reviewer 1 Report
In the manuscript “Quasi-Hyperbolically Symmetric $\gamma$-metric” the authors investigate the motion of particles in a quasi-hyperbolic version of the so-called $\gamma$-metric previously analysed by them (see Refs. [26-29]). The work aims at paving the way to eventually test the considered scenario against observations, namely those envisaging black holes shadows and relativistic jets.
The work is well written (some trimming should be in any case done, as for instance, Blandford is mistyped in Ref. [70]), and present relevant novel results. I am thus of the opinion that this work deserves publication in the journal Entropy.
The quality of the english is fine.
Author Response
See the attached reply

Reviewer 2 Report
Report on the paper :
Title :" Quasi–Hyperbolically Symmetric γ-Metric
" Authors: Luis Herrera, Alicia Di Prisco, Justo Ospino, Jaume Carot
This paper deals with the motion of test particles in the region inner to the naked singularity of a quasi–hyperbolically symmetric γ-metric. The geodesic equations are investigated and results are contrasted with the corresponding results obtained for the axially symmetric γ-metric, and the hyperbolically symmetric black hole. As in this latter case, it is found that test particles experience a repulsive force within the horizon (naked singularity), which prevents them to reach the center. However in the present case this behavior is affected by the parameter γ which measures the departure from the hyperbolical symmetry. These results are obtained for radially moving particles as well as for particles moving in the θ − r subspace. Possible relevance of these results in the explanation of extragalactic jets, is brought out.
Hyperbolically symmetric fluids are a type of fluid that exhibit hyperbolic symmetry, which is a type of mathematical symmetry associated with certain physical systems. In the context of fluids, hyperbolic symmetry refers to the fact that the fluid's velocity and pressure fields can be described using hyperbolic equations, which are partial differential equations that describe the behavior of physical systems in terms of wave-like properties. Hyperbolically symmetric fluids are often used to model complex fluid dynamics problems, such as shock waves, turbulence, and multi-phase flows. The hyperbolic symmetry of the fluid equations allows for the representation of nonlinear and discontinuous behavior in the fluid, making these equations well suited to describe a wide range of fluid dynamics problems. In general, the study of hyperbolically symmetric fluids is an area of active research in fluid dynamics, and advances in this field can have important implications for a wide range of industrial and scientific applications, such as energy production, aerospace engineering, and environmental science, among others.
The applications of the present study are clearly mentioned and studied. One of the most promising application of hyperbolical metrics are described. The modeling of extragalactic relativistic jet with an important issue in relativistic astrophysics, namely: modeling of the gravitational collapse of massive stars. The authors presented a novel model to analyze the physical properties of the hyperbolical version of the γ-metric. Therefore the endeavor of authors is praiseworthy and the presented models are certainly worth publishing. The physical description of the source is quite complete and I believe that the presented results are correct and will be of interest for people working on gravitational collapse.
Accordingly I recommend its publication as is.
Author Response
See the attached reply

Reviewer 3 Report
Quasi–hyperbolically symmetric $\gamma$-metric
In this paper the authors describe the motion of test particles in a quasi–hyperbolically symmetric $\gamma$-metric.
The authors found the geodesic equations and describe their properties in detail.
I consider the study is interesting and the manuscript is well written.
I would recommend, however, the authors to make a clear distinctions in the results between timelike and null geodesics. The description of null geodesics will support the claim that the results may be applicable to the recent EHT observations.
The authors should describe in detail (and supported with the equations of motion) how the repulsive effect in particles that they describe can be used to describe the jets present in some astrophysical black holes.
I consider the manuscript can be considered for publication after this issues are solved.
Author Response
See the attached reply

Round 2
Reviewer 3 Report
I consider the authors have solved the issues I raised properly and thus I consider the manuscript is now suitable for publication in Entropy.